# DRAFT-AND-TARGET SAMPLING FOR VIDEO GENERATION POLICY

## ABSTRACT

Video generation models have been used as a robot policy to predict the future states of executing a task conditioned on task description and observation. Previous works ignore their high computational cost and long inference time. To address this challenge, we propose Draft-and-Target Sampling, a novel speculative decoding-like inference paradigm for video generation policy that is training-free and can improve inference efficiency. We modify the classic principle of speculative decoding design and redefine the draft and target as two complementary denoising trajectories. To further speedup generation, we introduce token chunking and progressive acceptance strategy to reduce redundant computation. Experiments on three benchmarks show that our method can achieve up to 2.1x speedup and improve the efficiency of current state-of-the-art methods with minimal compromise to the success rate. Our code is available at anonymous github.

## 1 INTRODUCTION

Recent years have seen growing research interests in building an embodied agent with a *video generation policy* as its core (Du et al., 2023b; Hu et al., 2025). Video generation policy consists of two models, a video generation model and an action prediction model (Ko et al., 2024; Luo & Du, 2025). Unlike Vision Language Action (VLA) models (Kim et al., 2025) that directly map observations and instructions to actions and often overlook the underlying dynamics of the environment, video generation policies naturally model the progression of external states over time, they generate frames that capture the environment dynamics and implicitly contain actionable information on how to reach next visual state.

In building video generation policy, research efforts are either focused on action predictor or on video generation. To improve the action predictor, (Ko et al., 2024) reconstruct objects movements from depth maps and optical flows, whereas (Luo & Du, 2025) give attention on constructing a replay buffer to train the action predictor through self-supervised learning. Vision Language Models have been applied to assess and refine the quality of generated video content (Soni et al., 2024), while Vidman (Wen et al., 2024) introduces a dual stage pre-training on a large-scale robotics dataset to improve generalization on the downstream manipulation tasks.

Although previous works have achieved remarkable progress, little attention has been paid to the inference efficiency of video generation policies. The need for efficiency is critical in embodied agents, as a robot must be able to react in real time. Inspired by the success of speculative decoding (Leviathan et al., 2023), we propose a speculative decoding-like inference paradigm to reduce long inference time of video generation policy. In Large Language Model (LLM) acceleration, the basic idea of speculative decoding is to use a draft model to generate multiple draft tokens in a fast manner, then use a target model to verify these tokens in parallel. The success of this technique in LLM decoding is due to its idea to transform the decoding process from memory bandwidth bound to compute bound. However, this does not hold for most visual generation tasks as they generate high-resolution images, commonly 512×512 or higher, so the effect of speculative decoding is very small as they can easily reach the compute bound. On the other hand, in visual generation for robotics, the generated content usually has a relatively low resolution, often below 256×256, leaving room for exploring speculative decoding to accelerate inference.

In this research, we introduce Draft-and-Target Sampling (DTS) to speedup the inference of video generation policy. DTS does not need an independent draft model instead two complementary tra-

jectories from the same model. We consider the denoising trajectory as a collaboration between two sampling strategies on the same model. *Draft sampling* quickly jumps forward along the trajectory to generate a chunk of tokens, which is cheaper but may accumulate more errors. Then *target sampling* takes small steps to refine and verify this chunk in parallel to make sure the results are reliable. This collaboration lets a single model play both the draft and the target roles, which makes our inference framework simple and training-free. To further speedup inference and reduce redundant computation, we propose token chunking and a progressive acceptance strategy. In token chunking, we divide the entire denoising trajectories into several chunks and each chunk represents a shorter denoising trajectories. In the progressive acceptance strategy, we use L2 distance to measure the distribution between draft tokens and target tokens, and we allow a wider range of acceptance bound to replace the strict matching of this distribution in the original setting.

In summary, our main contributions are: **(1)** We propose a novel training-free speculative decoding-like framework for video generation policy, which combines two complementary denoising trajectories to accelerate inference and mitigate accumulated errors. **(2)** To further accelerate inference, we propose token chunking and progressive acceptance strategy, which divide the inference process into several chunks and progressively relax the acceptance bound, respectively. **(3)** Our inference framework achieves up to 2.1x speedup across three benchmarks. It achieves the state-of-the-art on iThor while resulting in a minimal compromise to success rate on Meta-World and Libero.

## 2 RELATED WORK

**Video Generation as Robot Planner** Video Generation Policy is a new paradigm that has been recently studied in the field of embodied intelligence (Zhou et al., 2024; Hu et al., 2025; Du et al., 2023a; Wu et al.; Black et al.; Guo et al., 2024; Chen et al., 2024; Ye et al., 2024; Bharadhwaj et al., 2024; Cheang et al., 2024). Unipi (Du et al., 2023b) was the first to formulate the video generation as policy in sequential decision making problem. It used a video generation model to synthesize videos about furture state and an inverse dynamics model to extract actions from generated videos. Building upon this, AVDC (Ko et al., 2024) designed a light-weighted video model and used optical flow to extract actions from videos. To further improve the performance of action model, GCP (Luo & Du, 2025) proposed to learn a diffusion policy to ground video models to action in an unsupervised manner. Another work, VideoAgent (Soni et al., 2024) utilized a vision language model to refine the visual trajectory, this approach achieved marginal improvement with significant time cost, which is unexpected in robot tasks. Our method used the video model itself to refine the visual trajectory in parallel, making the inference faster.

**Speculative Decoding** Speculative decoding is an inference paradigm that has been widely explored in large language models (Chen et al., 2023; Leviathan et al., 2023; Xia et al., 2022; 2024; Christopher et al., 2024; Spector & Re, 2023), which can speedup decoding by increasing the computation intensity to reach the GPU compute bound. Recent works have imporved this paradigm by scaling the paraller computation to further speedup generation (Cai et al., 2024; Li et al., 2024; Gao et al., 2025; Miao et al., 2023). Speculative decoding has also been applied to the domain of visual generation (De Bortoli et al.; Hu et al.), where some work has explored how to better relax the acceptance in order to solve the problem of token selection ambiguity (Jang et al., 2025; Park et al., 2025). Similar to our work, Spec-VLA (Wang et al., 2025) also applied speculative decoding to embodied intelligence, while they focused on vision language action models which takes action as tokens. And our work focused on diffusion-based video generation policy, in which the tokens are denoising trajectory. Moreover, Spec-VLA utilized the OpenVLA (Kim et al., 2025) as the target model and trained a draft model from scratch, while we made two complementary denoising trajectories within a single model to play a role of draft and target model, which makes our method training-free.

## 3 PRELIMINARIES

In this section, we first provide the formulation of video generation policy in Section 3.1. Then, in Section 3.2 we introduce the basic design of speculative decoding.

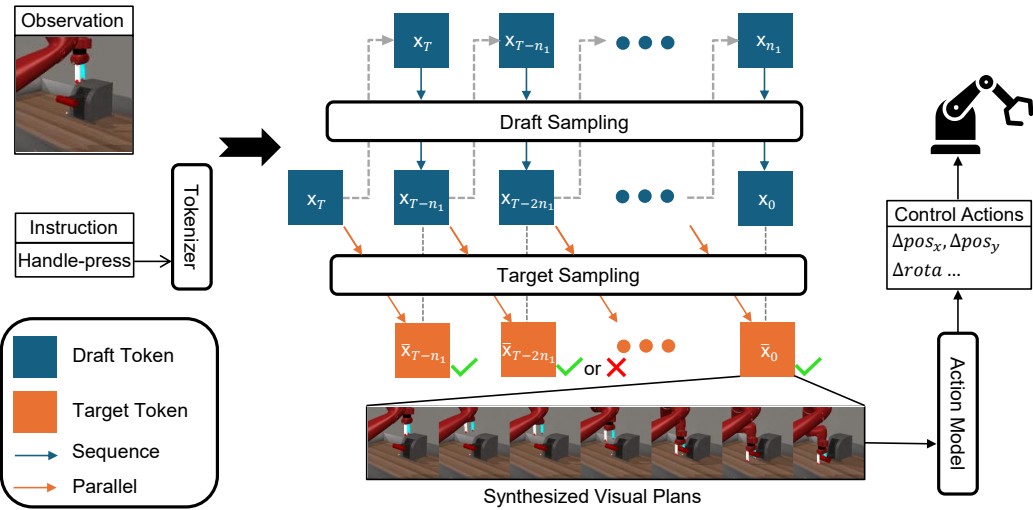

Figure 1: **The Draft-and-Target Sampling Framework.** In the first stage of DTS, the draft sampling generates a coarse denoising trajectory by taking large steps in a sequence manner, which provides a fast approximation of the denoising trajectory. Then the target sampling refines this trajectory in parallel by taking small steps, which leads to a more precise denoising trajectory. Finally, in verification stage we compare these two trajectories to determine if tokens are accepted or rejected. If a token is rejected, the draft sampling restarts from this position with its corresponding target token as the new starting point.

### 3.1 FORMULATION OF VIDEO GENERATION POLICY

Video Generation Policy is a type of embodied agent policy $P$, equipped with a video generation model at its core, conditioned on the observation $o$ and task instruction $i$. Compared to vision language action models $a = P(o, i)$ which directly map the observation $o$ and instruction $i$ to an action $a$, the video generation policy adopts a two-stage design. First, a video generation model generates a sequence of frames $V = \{v_1, v_2, \ldots, v_T\}$ that represent sub-goals of executing the task, given the inputs $o$ and $i$. Then, an action prediction model extracts a sequence of actions $A = \{a_1, a_2, \ldots, a_T\}$ that shows the agent how to complete the task from $V$.

### 3.2 SPECULATIVE DECODING FRAMEWORK

Speculative Decoding is a simple yet effective framework that was first proposed to accelerate LLM inference (Leviathan et al., 2023). In the draft sampling stage, it utilized a smaller and faster draft model $D_m$ to generates multiple draft tokens $\mathcal{D} = \{x_1, x_2, \ldots, x_n\}$ in a fast manner. In the verification stage, a target model $T_m$ computes the target distribution $\mathcal{G} = \{\bar{x}_1, \bar{x}_2, \ldots, \bar{x}_n\}$ of these draft tokens in parallel. Then we decide to accept or reject each draft token based on corresponding target token. Finally, during the resampling stage, the draft model resumes sampling from the first rejected token and continues until all tokens are accepted.

## 4 METHOD

### 4.1 DRAFT-AND-TARGET SAMPLING INFERENCE FRAMEWORK

A core idea behind original speculative decoding lies in using a draft model that can generate multiple tokens much faster than the target model, and a key challenge is how to design and train such a cheaper model that can balance generation speed and quality, which in practice requires empirical works in designing model architecture and a huge amount of GPU hours to train such a model.

For diffusion-based video generation models, we know that these models can generate high-quality denoising trajectories by running hundreds of continues denoising steps, and the DDIM solver can skip steps to significantly reduce inference time but struggle to generate reliably good trajectories when they jump too far because the issue of error accumulation. The interaction between the draft model and the target model in speculative decoding inspired us to combine these two complementary trajectories — a coarse but fast trajectory and a fine but slow trajectory — in a single inference framework. In our DTS inference framework, unlike most prior work that relies on independent draft and target models, we do not separate the draft model and target model, instead we separate the sampling strategy into two complementary roles. It means DTS only uses a single video generation model, while two sampling strategies: draft sampling and target sampling. The draft sampling stage is the same as the process of using a large-step DDIM solver. We start from a standard Gaussian noise $\epsilon \sim \mathcal{N}(0, I)$, the video generation model receives the visual feature from the visual encoder and the task feature from the tokenizer, together with the noise scheduling parameters $\alpha_t$. Instead of following the original fine-grained schedule, we take large steps of size $n_1$ to quickly generate the denoising trajectory. This produces a sequence of draft tokens $\mathcal{D}$ that approximate the denoising trajectory in a coarse but fast way. Formally, the diffusion iteration is given as:

$$\mathrm{x}_{t-n} = \sqrt{\alpha_{t-n}} \left( \frac{\mathrm{x}_t - \sqrt{1 - \alpha_t} \cdot \epsilon_\theta(\mathrm{x}_t, t)}{\sqrt{\alpha_t}} \right) + \sqrt{1 - \alpha_{t-n}} \cdot \epsilon_\theta(\mathrm{x}_t, t) \tag{1}$$

We define two different step sizes, where the draft trajectory uses a large step $n_1$ and the target trajectory uses a smaller step $n_2$:

$$n_2 < n_1 \tag{2}$$

The draft trajectory $\mathcal{D}$ is then sampled with step size $n_1$:

$$\{\mathrm{x}_{T-kn_1}\}, \qquad k = 0, 1, 2, ..., T/n_1 - 1 \tag{3}$$

where $\mathcal{D} = \{\mathrm{x}_T, \mathrm{x}_{T-n_1}, \mathrm{x}_{T-2n_1}, \ldots, \mathrm{x}_0\}$ denotes the draft trajectory obtained by skipping $n_1$ steps at a time and the first token $\mathrm{x}_T$ is the same as the initial noise $\epsilon$. $T$ is the maximum time step in the diffusion schedule. These draft tokens can be seen as a fast approximation of the full denoising trajectory and will later be verified in the target sampling stage.

During the target sampling stage, we first reuse the same initial Gaussian noise $\epsilon$ as $\mathrm{x}_T$ at the beginning of the draft sequence $\mathcal{D}$, while removing the final noiseless token $\mathrm{x}_0$. Then, the updated sequence $\mathcal{D}$, together with the visual feature and the task feature, is fed into the video model. The model revisits each position of the draft sequence in a progressive manner parallelly: for each token in $\mathcal{D}$, the video model refines it with a small step size $n_2$ along the denoising trajectory until it reaches the same timestep as its corresponding draft token, guided by the corresponding noise scheduling parameters $\alpha_t$. This process refines each draft token into a more accurate estimate of its true denoised state. There are $T/n_1$ sequences processed in parallel, for each draft token $\mathrm{x}_{T-kn_1}$, we refine it using smaller steps of size $n_2$ to obtain the target trajectory:

$$\{\bar{\mathrm{x}}_{(T-kn_1)-jn_2}\}, \qquad j = 0, 1, 2, ..., n_1/n_2 \text{ and } \bar{\mathrm{x}}_{T-kn_1} = \mathrm{x}_{T-kn_1} \tag{4}$$

Specifically, each draft token $\mathrm{x}_{T-kn_1}$ corresponds to one target sequence, which can be written as:

$$\mathcal{G}_k = \{\mathrm{x}_{(T-kn_1)}, \bar{\mathrm{x}}_{(T-kn_1)-n_2}, \bar{\mathrm{x}}_{(T-kn_1)-2n_2}, \ldots, \bar{\mathrm{x}}_{(T-(k+1)n_1)}\} \tag{5}$$

Finally, we get $\mathcal{G}_{all} = \{\bar{\mathrm{x}}_{T-n_1}, \bar{\mathrm{x}}_{T-2n_1}, \ldots, \bar{\mathrm{x}}_0\}$ which denotes the final target trajectory, and each target token is a better approximation of the corresponding draft token in $\mathcal{D}$. For instance, $\bar{\mathrm{x}}_{T-n_1}$ is a better approximation than $\mathrm{x}_{T-n_1}$, $\bar{\mathrm{x}}_0$ is a better approximation than $\mathrm{x}_0$.

$$\text{Accept } \mathrm{x}_t \text{ if } \mathrm{x}_t = \bar{\mathrm{x}}_t \text{ , else Resample } \mathrm{x}_t \tag{6}$$

By iteratively applying small step sampling, the target tokens accumulate less error compared with the draft tokens. Then, in the verification phase 6, we compute the distance between each pair of corresponding tokens in $\mathcal{D}$ and $\mathcal{G}$ and follow the strict matching to decide whether to accept or reject them. If the distance between two tokens is not $0$ then the draft token is rejected, the draft sampling restarts from the position of the first rejected token using the corresponding target token as a new starting point. For instance, if a comparison between $\bar{\mathrm{x}}_{T-2n_1}$ and $\mathrm{x}_{T-2n_1}$ leads to rejection, it indicates that the sequence sampled starting from $\mathrm{x}_{T-2n_1}$ (and any subsequent tokens derived from it) has an unacceptable error. Then the algorithm returns to using $\bar{\mathrm{x}}_{T-2n_1}$ as the starting point for sampling because $\bar{\mathrm{x}}_{T-2n_1}$ was derived from the accepted token $\mathrm{x}_{T-n_1}$. If all tokens are accepted, we directly return the last target token.

## 4.2 Challenges of Direct Application

When we directly apply the DTS inference framework to accelerate video generation policy, we observe two main problems: one is about compute bottleneck and error accumulation, the other is about high complexity of tokens.

**Compute Bottleneck and Error Accumulation** The implementation of target sampling relies on parallel computation of the GPU. As explained in Sec 4.1, the target sampling process the sequence $\mathcal{D}$ as a batch, and similar to Medusa (Cai et al., 2024) , the target sampling improves the compute-to-memory ratio by scaling the batch size of each forward pass. However, since GPU has its compute bound, if we keep scaling the batch size of a single forward pass, we reach the compute bound quickly, which makes the total forward time longer. At the same time, as $\mathcal{D}$ is generated by draft sampling with large step sizes, small errors gradually accumulate as the denoising process goes on, and this problem of error accumulation leads to a fact that if the draft sampling produces a long sequence of draft tokens, then the target sampling has to run the same batch size of tokens to obtain a sequence of more accurate target tokens. However, many tokens in the later positions are often rejected in the verification phase. This means time and computation is wasted and unnecessary.

**High Complexity of Tokens** Similar to the problem described in Spec-VLA (Wang et al., 2025), unlike LLMs which work on words, VLA models work on action tokens, they must generate robot control actions by understanding both the external environment and natural language instructions. These action tokens are naturally more complex and diverse than words. As for video generation policy, the tokens we work on are the denoising trajectory, the complexity and diversity of these tokens are even higher, the video model must understand the temporal and spatial motion patterns of the objects, the robot, and the environment dynamics, and then generate the visual trajectories of the objects and the robot. Moreover, the original speculative decoding setup requires the draft and target tokens to match strictly in order to guarantee the same distribution. Since the token we work on is denoising trajectory, achieving such strict alignment between two denoising trajectories is almost impossible, and it leads to frequent resampling and significant time consumption.

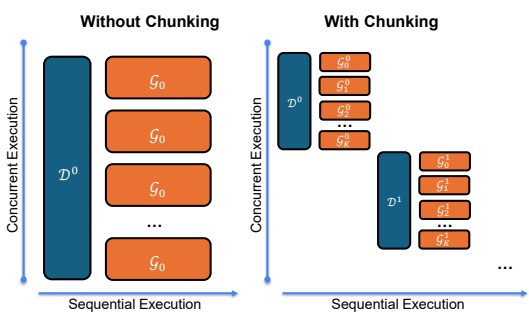

Figure 2: **Token Chunking.** In naive DTS, the draft sampling generates a long sequence in one pass, while token chunking divides the draft trajectory into smaller chunks that are processed chunk by chunk.

## 4.3 Token Chunking and Progressive Acceptance Strategy

To address these challenges, we design two strategies to make DTS inference framework more efficient for video generation policy.

**Token Chunking** We first introduce a token chunking strategy to solve the problem of compute bottleneck and error accumulation. As explained in Sec 4.2, this problem is caused by too many draft tokens being sampled at one time during the draft sampling stage. We therefore divide the process into smaller chunks instead of letting the draft sampling stage produce a long sequence of draft tokens in one step. Each chunk is verified with corresponding target tokens before moving to the next, which reduces the time that later tokens are rejected due to errors propagated from earlier steps. This design not only reduce time and computational costs but also can reduce accumulated error like target sampling, as each chunk returns the last target token when it ends, which then serves as the starting point for the next draft sampling stage, rather than the previous draft token. Formally, draft sampling no longer generates the entire sequence $\mathcal{D}$ as in Eq. 3 at a time. Instead, it generates tokens in smaller chunks identified by an index $i$, each of which contains $L$ tokens:

$$\mathcal{D}^i = \{\mathrm{x}_{T-(k+iL)n_1}\}, \qquad k = 0, 1, 2, ..., L-1, \ \text{ and } \ i = 0, 1, ..., n_1/L \tag{7}$$

where $\mathcal{D}^i$ denotes the $i$-th draft chunk, $L$ is the chunk length, and the entire sequence $\mathcal{D} = \{\mathcal{D}^0, \mathcal{D}^1, \ldots, \mathcal{D}^{n_1/L}\}$. For each draft chunk $\mathcal{D}^i$, the target sampling generates k corresponding

sequence in parallel:

$$\mathcal{G}_k^i = \{\bar{x}_{(T-(k+iL)n_1)-jn_2}\}, \qquad j = 0, 1, 2, ..., n_1/n_2 \tag{8}$$

And each draft token $x_{T-(k+iL)n_1}$ corresponds to one target sequence, which can be written as:

$$\mathcal{G}_k^i = \{x_{T-(k+iL)n_1}, \bar{x}_{(T-(k+iL)n_1)-n_2}, \bar{x}_{(T-(k+iL)n_1)-2n_2}, \ldots, \bar{x}_{T-(k+1+iL)n_1}\} \tag{9}$$

where $\mathcal{G}^i$ refines the tokens in $\mathcal{D}^i$, and the final sequence $\mathcal{G}_{all} = \{\mathcal{G}^0, \mathcal{G}^1, \ldots, \mathcal{G}^{n_1/L}\}$. The target sampling is responsible for refining the draft tokens within the current chunk $\mathcal{D}^i$. And the final target token of each chunk becomes the starting point for the next chunk. We visualize this strategy as shown in the figure 2.

**Progressive Acceptance Strategy** We introduce a progressive acceptance strategy to gradually relax the matching threshold instead of the original strict setting. Let's first explain how to calculate the distance between draft token $x_t$ and target token $\bar{x}_t$. During the verification stage, $x_t$ and $\bar{x}_t$ can be consider as two different approximations of the same point $t$ along the denoising trajectory. Their distance is naturally close, so we directly use L2 distance to measure this proximity. We choose L2 here because it is easier to amplify cumulative deviations across dimensions, unlike other methods such as MSE which tend to spread errors evenly over all dimensions and dilute the overall difference. In the progressive acceptance strategy, we allow to accept draft token $x_t$ within a wider range instead of the strict match, and this range expands as the chunk moves. Specifically, we introduce the matching threshold $\pi$ to decide whether to accept or reject a token, and another hyperparameter $\Delta\pi$ that controls the increment of $\pi$ at each round. This setup expands the token acceptance region to be a more flexible range instead of the strict match between draft token $x_t$ and target token $\bar{x}_t$.

$$\text{Accept } x_t \text{ if } \|x_t - \bar{x}_t\|_2 \leq \pi_k, \quad \pi_{k+1} = \pi_k + \Delta\pi \tag{10}$$

In this way, the algorithm allows for minor deviations in local while keeping the overall trajectory consistent. This flexibility is crucial for efficiency, where perfect alignment between such two generated sequences is almost impossible. By combining token chunking and progressive acceptance strategy, the overall DTS framework achieves a balance between efficiency and accuracy. It avoids token waste without sacrificing the generation quality.

## 5 EXPERIMENTS

We evaluate and analyze the proposed DTS framework and two components in this section. In Section 5.1, we first introduce the baselines and benchmarks. In Section 5.2, we compare DTS framework to the baselines across two robot manipulation task and one navigation task on simulator. Finally, in Section 5.3, we conduct the ablation studies to show the importance of token chunking and progressive acceptance strategy. The experiments environment is summarized in Appendix B.

### 5.1 EXPERIMENT SETUP

**Baselines** We conduct experiments on two state-of-the-art video generation policies and compare their performance across three benchmarks to validate the effect of our DTS framework. The first one is AVDC (Ko et al., 2024), it utilizes a text-conditioned video diffusion model as the robot visual planner to generate a video represented a sequence of sub-goals and optical flow prediction as the action predictor to extract robot control actions from the generated video. Another baseline is GCP (Luo & Du, 2025), it utilizes the same video diffusion model as AVDC, while it trains a diffusion policy as the action predictor which can achieve a better alignment between generated video and actions.

**Benchmarks** Following these baselines, we evaluate our DTS framework on three widely used benchmarks, including two robot manipulation tasks (Libero and Meta-World) and one navigation task (iThor). Libero (Liu et al., 2023) is a benchmark for life-long robot learning, which contains diverse household tasks. Meta-World (Yu et al., 2020) is a benchmark which includes a wide range of short-horizon tasks for evaluating the robot policy. iThor (Kolve et al., 2017) is a simulator for visual navigation, in which the embodied agents are asked to navigate to a given object in room scenarios. We provide details of benchmarks in Appendix C. Since the released pretrained model of GCP only covers Libero, and AVDC releases their pretrained model on Meta-World and iThor, we align our experiment setup with these resources and fix the random seed to make the experiments reproducible.

Table 1: **Comparison results on 4 types of rooms in iThor.** We reproduce AVDC with 100 denoising steps as in the original paper as our baseline, while AVDC-10 is a faster variant that uses only 10 denoising steps. * denotes the reported results from the original paper.

| Method | Kitchen | Living Room | Bedroom | Bathroom | **Overall** | **Speedup** |
|---|---|---|---|---|---|---|
| BC-Scratch* | 1.7 | 3.3 | 1.7 | 1.7 | 2.1 | / |
| BC-R3M* | 0.0 | 0.0 | 1.7 | 0.0 | 0.4 | / |
| AVDC* | 26.7 | 23.3 | 38.3 | 36.7 | 31.3 | 1x |
| **AVDC** | 30 | 16.7 | 35 | 26.7 | 27.1 | 1x |
| **AVDC-10** | 25 | 11.7 | 30 | 21.7 | 22.1 | 8.8358x |
| **+ Ours** | 30 | 25 | 38.3 | 23.3 | 29.15 | 2.1445x |

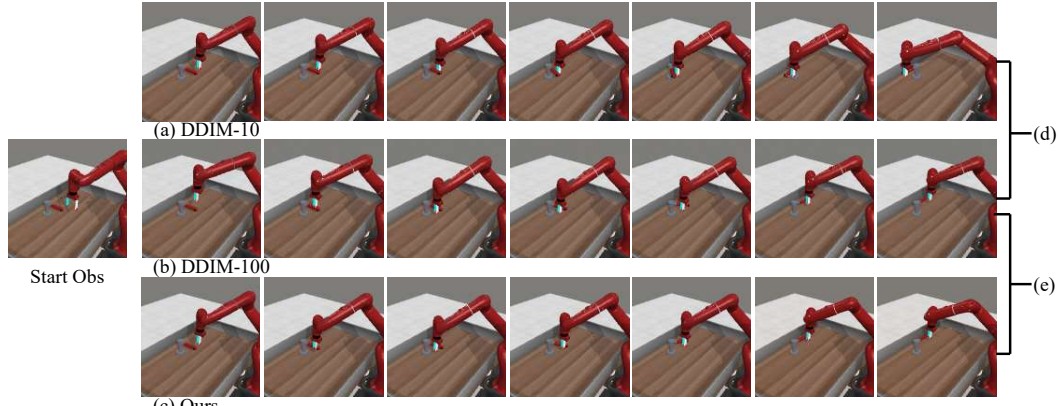

Figure 3: **Meta-World: faucet-close.** In this task, the agent is required to rotate the faucet handle and close it. As shown in comparison (d), the trajectory generated by DDIM-10 is markedly different from DDIM-100, with the faucet being rotated in an entirely different direction. As shown in comparison (e), our trajectory closely align with DDIM-100, both successfully closing the faucet.

## 5.2 OVERALL RESULT

**iThor** We compare AVDC with and without our DTS inference framework on 12 object navigation tasks across in iThor, which are distributed across four different types of rooms, and we run 20 trials for each task. We follow the original paper setting and reproduce AVDC with the DDIM solver using 100 denoising steps, and we also show AVDC with only 10 denoising steps as a fast but less accurate variant, the resolution of all the generated frames is (64 x 64). For the DTS inference framework, we set the step size $n_1$ to 10 and $n_2$ to 2, the length of a chunk as 6, the matching threshold $\pi$ as 11, and $\Delta\pi$ as 1.5. In Table 1, we can see that AVDC with 100 denoising steps achieves an overall mean success rate of 27.1% across 4 rooms, with an average runtime of 3.013s. The AVDC with 10 denoising steps tasks only 0.341s to generate a video but drops in performance, reaching only 22.1% overall success rate. Surprisingly, the AVDC equipped with our DTS framework achieves a much better balance between efficiency and success rate, since it reaches an overall success rate of 29.15% with an average runtime of 1.405s, which means a 2.14x speedup over the baseline while also improving the success rate by 2.05%. We also evaluate different chunk lengths and matching thresholds to show the robustness of our method, as shown in Section 5.3. We also report the results of two baselines used in the original paper (Ko et al., 2024). BC-Scratch is a multi-task behavioral cloning method that uses a ResNet-18 (He et al., 2016) trained from scratch as its visual encoder. BC-R3M uses the R3M model (Nair et al., 2022), which was pre-trained on robot manipulation tasks and it performs worse on navigation tasks.

**Meta-World** We conduct experiments on 11 tasks in Meta-World to compare AVDC with and without our DTS inference framework, and we follow the original paper setting, we reproduce AVDC with its released model using the DDIM solver and run 100 denoising steps. The resolution of all the

Table 2: **Comparison results on 11 tasks in Meta-World.** We reproduce AVDC with 100 denoising steps as in the original paper as our baseline, while AVDC-10 is a faster variant that uses only 10 denoising steps. Following the original setting, we evaluate each task for 25 trials with 3 camera poses, 825 in total. * denotes the reported results from the original paper.

| Task | BC-Scratch* | BC-R3M* | AVDC* | **AVDC** | **AVDC-10** | **+ Ours** |
|---|---|---|---|---|---|---|
| door-open | 21.3 | 1.3 | 72 | 72 | 69.3 | 69.3 |
| door-close | 36 | 58.7 | 89.3 | 97.3 | 100 | 97.3 |
| basketball | 0.0 | 0.0 | 37.3 | 33.3 | 16 | 26.6 |
| shelf-place | 0.0 | 0.0 | 18.7 | 21.3 | 5.3 | 4 |
| btn-press | 34.7 | 36 | 60 | 52 | 52 | 45.3 |
| btn-press-top | 12 | 4 | 24 | 14.7 | 4 | 18.6 |
| faucet-close | 18.7 | 18.7 | 53.3 | 49.3 | 50.6 | 48 |
| faucet-open | 17.3 | 22.7 | 24 | 30.7 | 32 | 29.3 |
| handle-press | 37.3 | 28.0 | 81.3 | 81.3 | 79.9 | 84 |
| hammer | 0.0 | 0.0 | 8 | 9.3 | 24 | 26.7 |
| assembly | 1.3 | 0.0 | 6.7 | 5.3 | 2.6 | 4 |
| **Overall** | 16.2 | 15.4 | 43.1 | 42.4 | 38.3 | 41.2 |
| **Speedup** | / | / | 1x | 1x | 7.4814x | 1.3487x |

generated frames is (128 x 128). For our DTS inference framework, we set the step size $n_1$ to 10 and $n_2$ to 2, the length of a chunk as 10, the matching threshold $\pi$ as 11, and $\Delta\pi$ as 1.5, which means that in the first run, the draft sampling generates a complete denoising trajectory of ten tokens, then we send these tokens to target sampling for verification, and if any token is rejected then the next draft sampling generates the denoising trajectory starting from the rejection position until the target frame, which is the final noise-free one. Since our DTS framework can be regarded as a refinement to correct the accumulated error of AVDC with the DDIM solver using only ten denoising steps, we also show the results of AVDC with 10 denoising steps in Table 2. For each task, the policy is evaluated with three camera poses and 25 trials. On all 11 tasks, our baseline AVDC with 100 denoising steps takes an average of 6.03s and achieves an overall mean success rate of 42.4%, while AVDC with ten denoising steps takes only 0.806s on average but the success rate drops to 38.3%, which is similar to the original paper. Our DTS inference framework achieves a balance between efficiency and accuracy, since it speeds up AVDC by about 1.3487x with only 1.2% loss in success rate, reaching an overall success rate of 41.2%. Different chunk length can lead to even faster acceleration, as discussed in Sec 5.3. We visualize several tasks from iThor and Meta-World to show the difference among three generated trajectories in Appendix E. Here we take the faucet-close task as an example in Figure 3, it can be seen that the pose and rotation of the robotic arm in our trajectory is closer to the DDIM with 100 denoising steps.

**Libero** We compare the performance of GCP with and without our DTS inference framework on 8 manipulation tasks in Libero, as well as the sampling time of the video model. The resolution of all the generated frames is (128 x 128). We run 20 trials for each task to reproduce GCP using DDPM and DDIM solvers with 100 denoising steps and to evaluate our method. In our setting, we set the step size $n_1$ to 10 and $n_2$ to 2, the length of a chunk as 6, the matching threshold $\pi$ as 10, and $\Delta\pi$ as 1.5, which means that each draft sampling step generates a sequence of 6 draft tokens, and the target sampling verifies a batch of 6 tokens in parallel. For each chunk, the acceptance or rejection of a token is determined by a progressively increasing threshold that starts at 10 and grows by 1.5 with each subsequent round. See Table 3 for the overall results. The experiments show that our DTS inference framework is effective for video generation policy, since it enables GCP to achieve about 1.6x speedup without compromise to the success rate, and we also try different numbers of chunk length to show the robustness to hyperparameters, this can be seen in Section 5.3.

## 5.3 ABLATION STUDY

In this section, we conduct ablation studies to analyze the contribution of token chunking and progressive acceptance strategy to the DTS inference framework. We focus on the effect of different chunk lengths and matching thresholds on efficiency and success rate.

Table 3: **Comparison results on 8 tasks across two different scenes in Libero.** GCP* denotes the baseline method goal-conditioned policy, BC* is a multi-task behavioral cloning method, and the reported results are directly taken from the original paper. GCP-P and GCP-I represent reproduced results using DDPM and DDIM samplers, respectively. The success rates are in increments of 5% because each task is evaluated over 20 trials, leading to discrete averages.

| Task | BC* | GCP* | GCP-P | GCP-I | +Ours |
|------|-----|------|-------|-------|-------|
| put-red-mug-left | 8.8±5.3 | 38.4±15.3 | **45** | 30 | 30 |
| put-red-mug-right | 15.2±7.8 | 40.8±7.8 | 40 | 40 | **40** |
| put-white-mug-left | 32.0±12.9 | 51.2±3.9 | 55 | 40 | **70** |
| put-Y/W-mug-right | 21.6±12.3 | 38.4±8.6 | 30 | **40** | 25 |
| put-choc-left | 19.2±9.3 | 70.4±12.8 | **70** | 50 | 65 |
| put-choc-right | 12.8±9.3 | 79.2±3.9 | 80 | 70 | **90** |
| put-red-mug-plate | 7.2±5.3 | 72.8±6.4 | 65 | **70** | 65 |
| put-white-mug-plate | 20.0±11.3 | 25.6±11.5 | 15 | 15 | **15** |
| Overall | 17.1±9.2 | 52.1±8.8 | 50 | 44.375 | **50** |
| Time cost(s) | / | / | 10.8335 | 10.8089 | **6.7578** |
| Speedup | / | / | 1x | 1.002x | **1.6031x** |

**Token Chunking** We first analyze the effect of token chunking on the efficiency and success rate. In this experiment, we fix the matching threshold with an increment of 1.5 and compare different chunk lengths on three benchmarks. The result on three benchmarks is summarized in Appendix D.1. From the results, we can observe a clear pattern on efficiency. Smaller chunks, such as 2 or 4 tokens, consistently lead to faster inference, often achieving the highest speedup across all settings. In contrast, larger chunks, such as 8 or 10 tokens, tend to produce lower acceleration, since the target sampling needs to handle longer draft sequences at once, which reduces efficiency. *A medium chunk length, especially 6 tokens, consistently shows the best balance across all three benchmarks.* In Meta-World, Chunk-6 improves the success rate compared to shorter or longer chunks while still achieving 1.47x speedup. In iThor, Chunk-6 gives a strong gain by reaching about 2.14x acceleration and even higher success rates compared to the baseline. In Libero, Chunk-6 achieves around 2x acceleration and the highest average success among all chunk length. These findings indicate that while chunk length directly controls the efficiency of our DTS framework, its effect on success rate is less predictable. In practice, chunk length 6 appears to be a robust choice, as it avoids severe performance drops while still providing expected speedup.

**Progressive Acceptance Strategy** We further study the effect of the Progressive Acceptance Strategy, which accepts draft tokens under different matching thresholds with a fix chunk length of 6. The results on Libero are shown in Appendix D.2. From the results we can observe a consistent trend regarding efficiency. *Lower thresholds usually result in higher computation cost*, because they require the draft tokens and target tokens to have a stricter distribution match, so draft tokens are more likely to be rejected and need to be resampled. On the other hand, *higher thresholds allow a more relaxed acceptance*, and this directly leads to faster acceleration. However, threshold 8 gives slower speedup than threshold 6. At the same time, when the threshold is set to 10, our method finishes inference in only 5.43s, which means that in time-sensitive tasks we can speed up inference by around 2x compared to the baseline while losing only 1.2% in success rate.

## 6    CONCLUSIONS

We introduce a training-free inference framework inspired by speculative decoding. It combines a coarse but fast trajectory to approach the denoising path and a fine but slow trajectory to refine it without having to train a separate draft model. In addition, we propose token chunking and progressive acceptance strategy to reduce redundant computation. Experiments show that our method can significantly accelerate the inference with minimal compromise to the success rate. We believe our method can inspire more efficient inference designs for diffusion-based models and broader robotic applications.

ETHICS STATEMENT

This work focuses on developing a training-free inference framework to accelerate video generation policy. It does not involve human subjects, sensitive personal data, or applications with immediate societal risks. The datasets used in our experiments are publicly available and widely used in prior research. We have carefully followed the ICLR Code of Ethics throughout the research and writing process, and we believe our work has no ethical concerns.

REPRODUCIBILITY STATEMENT

To make sure this work can be reproduced, we attach the code and corresponding guildline at anonymous github. In addition, we save the entire environment in the cloud server and can share it after the paper is accepted.

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

APPENDIX

## A    USE OF LLMs

In this work, we only used LLM to help polish the writing of this paper. LLM did not involve in designing the method, conducting the experiments, or analyzing the results. All technical contributions and experimental findings are original to us authors.

## B    DETAILS OF EXPERIMENTAL ENVIRONMENT

We conduct experiments on two different workstations. For iThor and Meta-World, we conduct the experiments on a WSL environment running Ubuntu 22.04 with a 4090 GPU. For Libero, we use a native Ubuntu 22.04 environment with another 4090 GPU.

## C    DETAILS OF BENCHMARKS

**iThor** (Kolve et al., 2017) is a 2D simulation environment for embodied visual navigation and common sense reasoning. In our experiments, the agent is randomly started in a room and asked to navigate to a target object, such as a toaster or television. At each step, the agent can move forward, rotate left, rotate right, or declare the episode done. We select 12 objects distributed across four room types, with three tasks per room. A trajectory is considered successful if the agent reaches within 1.5 meters of the target object and has it in view before the maximum number of steps is reached or the agent signals completion.

**Meta-World** (Yu et al., 2020) provides a set of short-horizon robot manipulation tasks performed by a Sawyer robotic arm. In our study, we evaluate our method on 11 tasks with three fixed camera angles. During evaluation, the starting positions of the robot and the objects are randomized according to the seed. A rollout is considered successful if the agent reaches a position very close to the goal state. We run 25 seeds per camera angle and aggregate the results to compare baselines and our method.

**Libero** (Liu et al., 2023) is a benchmark for life-long robot learning and contains a diverse set of household manipulation tasks. In our experiments, we follow the original evaluation setup and compare baselines and our method on 8 tasks. A task is considered successful if the target object reaches a range of predefined goal state. If a success is detected during the rollout, the episode ends immediately and counts as a successful rollout. Otherwise, if all synthesized video are executed without reaching the goal, the episode is counted as a failure.

## D    ABLATION STUDIES

### D.1    DIFFERENT CHUNK LENGTH

In this section, we present the ablation results of different chunk lengths on three benchmarks: iThor, Meta-World, and Libero. We set the chunk length to 2, 4, 6, 8, and 10 tokens.

Table 4: **Ablation on 4 types of rooms in iThor.** We evaulate different chunk length.

| Method | Kitchen | Living Room | Bedroom | Bathroom | Overall | Speedup |
|--------|---------|-------------|---------|----------|---------|---------|
| Chunk-2 | 30 | 20 | 31.7 | 18.3 | 25 | 1.934x |
| Chunk-4 | 28.3 | 16.7 | 31.7 | 28.3 | 26.25 | 2.1107x |
| Chunk-6 | 26.7 | 23.3 | 38.3 | 36.7 | 31.3 | 2.1445x |
| Chunk-8 | 28.3 | 15 | 26.7 | 21.7 | 22.9 | 2.0655x |
| Chunk-10 | 16.7 | 30 | 33.3 | 31.7 | 27.9 | 1.9935x |

Table 5: **Ablation on 11 tasks in Meta-World.** We evaluate different chunk lengths.

| Task | Chunk-2 | Chunk-4 | Chunk-6 | Chunk-8 | Chunk-10 |
|---|---|---|---|---|---|
| door-open | 60 | 68 | 70.7 | 64 | 69.3 |
| door-close | 97.3 | 98.7 | 97.3 | 100 | 97.3 |
| basketball | 14.7 | 13.3 | 21.3 | 22.7 | 26.6 |
| shelf-place | 1.3 | 1.3 | 1.3 | 0 | 4 |
| btn-press | 53 | 60 | 50.7 | 42.7 | 45.3 |
| btn-press-top | 2.7 | 8 | 5.3 | 0 | 18.6 |
| faucet-close | 44 | 49.3 | 45.3 | 42.7 | 48 |
| faucet-open | 33 | 32 | 30.7 | 32 | 29.3 |
| handle-press | 76 | 76 | 84 | 76 | 84 |
| hammer | 29.3 | 19.9 | 21.3 | 24 | 26.7 |
| assembly | 4 | 2.7 | 8 | 4 | 4 |
| Overall | 37.8 | 39 | 39.6 | 37.1 | 41.2 |
| Speedup | 1.5351x | 1.5944x | 1.475x | 1.4747x | 1.3487x |

Table 6: **Ablation on 8 tasks across two different scenes in Libero.** We evaulate different chunk length.

| Task | Chunk-2 | Chunk-4 | Chunk-6 | Chunk-8 | Chunk-10 |
|---|---|---|---|---|---|
| put-red-mug-left | 25 | 25 | 40 | 30 | 35 |
| put-red-mug-right | 45 | 45 | 50 | 55 | 40 |
| put-white-mug-left | 40 | 60 | 60 | 45 | 50 |
| put-Y/W-mug-right | 25 | 25 | 30 | 30 | 15 |
| Overall | 33.75 | 38.75 | 45 | 40 | 35 |
| Time cost(s) | 5.5872 | 4.9862 | 5.4128 | 5.6836 | 6.8004 |
| Speedup | 1.9393x | 2.173x | 2.002x | 1.9064x | 1.5933x |

| Task | Chunk-2 | Chunk-4 | Chunk-6 | Chunk-8 | Chunk-10 |
|---|---|---|---|---|---|
| put-choc-left | 55 | 65 | 70 | 50 | 60 |
| put-choc-right | 70 | 60 | 70 | 65 | 80 |
| put-red-mug-plate | 60 | 65 | 60 | 55 | 50 |
| put-white-mug-plate | 10 | 25 | 10 | 5 | 15 |
| Overall | 48.75 | 53.75 | 52.5 | 43.75 | 51.25 |
| Time cost(s) | 5.5968 | 5.0231 | 5.4558 | 5.7299 | 6.8162 |
| Speedup | 1.9354x | 2.1564x | 1.985x | 1.8904x | 1.5891x |

## D.2 DIFFERENT MATCHING THRESHOLD

In this section, we present the ablation results of different matching thresholds on iThor and Libero. We set the matching threshold to 4, 6, 8, 10, and 12.

Table 7: **Ablation on 4 types of rooms in iThor.** We evaulate different matching threshold.

| Method | Kitchen | Living Room | Bedroom | Bathroom | Overall | Speedup |
|---|---|---|---|---|---|---|
| Thre-4 | 21.7 | 23.3 | 15 | 23.3 | 20.8 | 1.8074x |
| Thre-6 | 31.7 | 20 | 38.3 | 23.3 | 28.3 | 1.8831x |
| Thre-8 | 30 | 20 | 36.7 | 21.7 | 27.1 | 1.8962x |
| Thre-10 | 30 | 23.3 | 33.3 | 16.7 | 25.8 | 1.8691x |
| Thre-12 | 25 | 23.3 | 33.3 | 20 | 25.4 | 1.889x |

Table 8: **Ablation on 8 tasks across two different scenes in Libero.** We evaulate different matching threshold.

| Task | Thre-4 | Thre-6 | Thre-8 | Thre-10 | Thre-12 |
|------|--------|--------|--------|---------|---------|
| put-red-mug-left | 45 | 35 | 30 | 30 | 40 |
| put-red-mug-right | 45 | 40 | 40 | 40 | 50 |
| put-white-mug-left | 55 | 35 | 50 | 70 | 60 |
| put-Y/W-mug-right | 20 | 20 | 20 | 25 | 30 |
| put-choc-left | 45 | 70 | 50 | 65 | 70 |
| put-choc-right | 60 | 75 | 60 | 90 | 70 |
| put-red-mug-plate | 65 | 65 | 70 | 65 | 60 |
| put-white-mug-plate | 10 | 20 | 5 | 15 | 10 |
| Overall | 43.125 | 45 | 40.625 | 50 | 48.75 |
| Time cost(s) | 9.6118 | 7.1290 | 8.2853 | 6.7578 | 5.4343 |
| Speedup | 1.1271x | 1.5196x | 1.3076x | 1.6031x | 1.994x |

# E  VISUALIZATION

In this section, we provide the visualization of different tasks from the Meta-World and iThor benchmark, comparing three approaches: DDIM-10, DDIM-100, and our method. And we analyze three trajectories and the reasons for failure or success in the caption, more demo can be seen in the supplementary material.

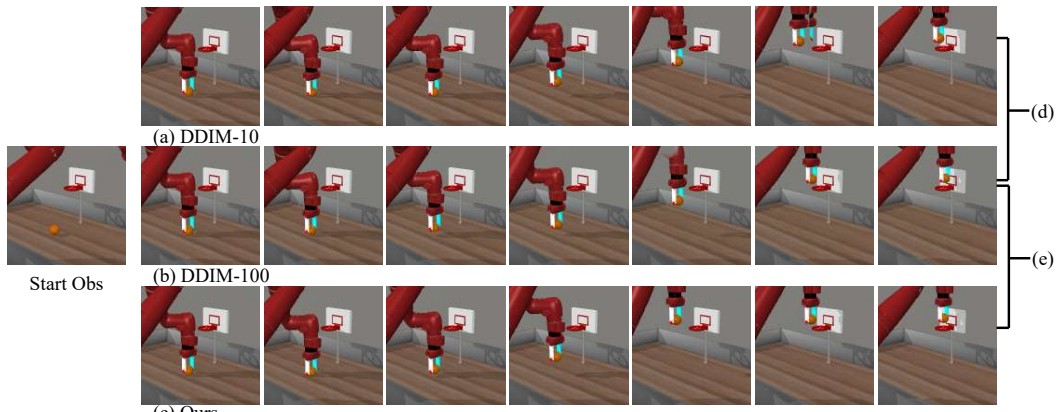

Figure 4: **Meta-World: Basketball.** In this task, the agent needs to place the basketball into the basket. As shown in comparison (d), although the trajectory generated by DDIM-10 is close to DDIM-100, a ghosting artifact appears in the second-to-last frame where an extra fragment of the robotic arm is generated near the true arm, and in the final frame the basketball is positioned slightly off-center relative to the basket, which leads to fail in this task. In contrast, our trajectory closely aligned with DDIM-100, both successfully placing the basketball directly above the center of the basket.

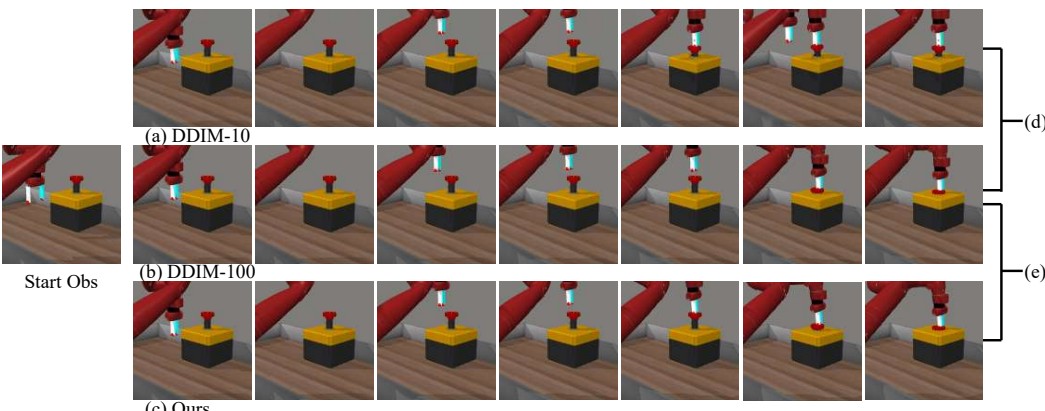

Figure 5: **Meta-World: Button-press-topdown.** In this task, the agent is required to press the button from a top-down direction. As shown in comparison (d), although the trajectory generated by DDIM-10 is close to DDIM-100, a ghosting artifact appears in the second-to-last frame where an extra fragment of the robotic arm is generated near the true arm, and in the final framethe button not completely pushed down. In contrast, our trajectory closely aligned with DDIM-100, both successfully press the button all the way down.

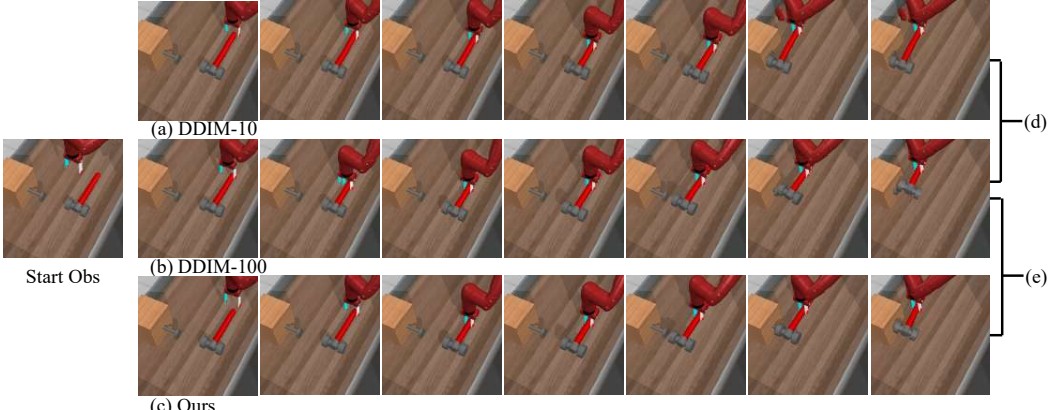

Figure 6: **Meta-World: Hammer.** In this task, the agent is required to use a hammer to drive a nail into the wooden block. As shown in the figure, the trajectory generated by DDIM-10 is different from DDIM-100, losing track of it in the final two frames and failing to drive the nail into the block. In contrast, our trajectory closely match DDIM-100, with both successfully embedding the nail into the block.

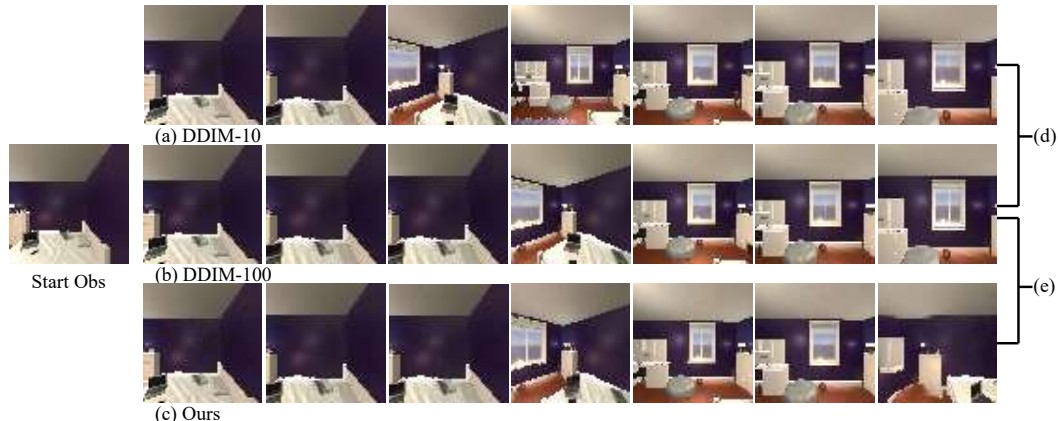

Figure 7: **iTHOR: desk-lamp.** In this task, the agent starts in a bedroom and must find the desk lamp among other furniture. As shown in the figure, both DDIM-10 and DDIM-100 fail to locate the desk lamp. In contrast, our method successfully finds the desk lamp and navigates toward it beyond the final frame.

