# OpenReview forum: "Draft-and-Target Sampling for Video Generation Policy"
_ICLR.cc/2026/Conference — Submitted to ICLR 2026_

### Official Review · Reviewer_pqge · 2025-10-29

**Soundness:** 3
**Presentation:** 2
**Contribution:** 2
**Rating:** 2
**Confidence:** 3

**Summary:**

This paper proposes Draft-and-Target Sampling (DTS), a speculative decoding method for video generation policies. The approach employs large diffusion steps for the draft phase and smaller diffusion steps for the acceptance/rejection phase. To mitigate accumulation errors and improve efficiency, the paper introduces token chunking and progressive acceptance strategies. Experiments conducted on iTHOR, MetaWorld, and LIBERO demonstrate improved inference efficiency while maintaining comparable policy performance.

**Strengths:**

1. The paper presents a novel perspective on video generation policies, focusing on improving inference efficiency. To the best of my knowledge, the proposed strategy of combining large and small diffusion steps for speculative decoding has not been explored before.
2. The experimental evaluation is comprehensive, covering three distinct domains, and the results show consistent and satisfactory performance.

**Weaknesses:**

1. **Lack of theoretical analysis of acceleration**:
   While the empirical study is extensive, the paper lacks theoretical discussion or quantitative analysis of the acceleration achieved:
   - The assumption in Line 50 that video generation policies usually have low resolutions is questionable. Recent works such as Vidar [1] demonstrate a clear trend toward higher resolutions in this paradigm. The paper does not analyze how the proposed method is applicable under different resolutions (e.g., whether it is memory-bound or compute-bound).
   - Although the paper presents detailed algorithmic formulations, it omits an analysis of time complexity, particularly regarding the impact of token chunking. This makes it difficult for readers to understand the expected acceleration ratio or to choose suitable hyperparameters
   - If the authors claim that generation is memory-bound, then the memory cost of token chunking (e.g., the number of model parameter loads) should be examined, as it may remain similar or even increase with chunking.
   - The sequential nature of token chunking may limit parallelism. This trade-off should be discussed explicitly.
2. **Presentation and clarity issues**:
   - Section 4 is overly verbose, containing many complex formulas. A clear schematic figure illustrating the overall process would greatly enhance readability.
   - The motivation for token chunking as a way to mitigate accumulation error of draft sampling is somewhat trivial and could be summarized more concisely.
   - The discussion in Section 4.3 about token chunking reducing accumulated error of target sampling (Line 262) is insightful but should be introduced earlier, e.g., in Section 4.2.
   - The motivation of the Progressive Acceptance Strategy is not sufficiently explained or justified.
   - There is considerable repetition across the three benchmark descriptions in Section 5.2 (e.g., shared hyperparameter settings), which could be consolidated to improve conciseness.

**Questions:**

1. What is the motivation and intuition behind the Progressive Acceptance Strategy?
2. How does this work compare to Accelerated Diffusion Models via Speculative Sampling [2], which also avoids training a separate draft model?
3. Why is DDIM-10 chosen as the baseline solver for diffusion sampling instead of more advanced solvers such as DPM-Solver?
4. Could the proposed approach be extended to general image or video generation models, beyond policy-oriented settings?
5. On iTHOR, why does DTS improve policy performance in addition to acceleration? Were the results averaged over multiple runs to ensure statistical significance?

[1] Vidar: Embodied Video Diffusion Model for Generalist Manipulation

[2] Accelerated Diffusion Models via Speculative Sampling.

---

> ### Author Response · Authors · 2025-11-21
> **Response to Reviewer pqge (1/3)**
>
> We thank reviewer for their careful and constructive review. Please also see our response to each of the questions below.
>
> W1
>
> >The assumption in Line 50 that video generation policies usually have low resolutions is questionable. Recent works such as Vidar [1] demonstrate a clear trend toward higher resolutions in this paradigm. The paper does not analyze how the proposed method is applicable under different resolutions (e.g., whether it is memory-bound or compute-bound).
>
> We thank the reviewer for this suggestion. In our experiments, we evaluate two different resolutions: Libero and Meta-World at 128×128, and iThor at 64×64. In both settings, our method achieves up to 2.1× acceleration, which shows that DTS is effective across a range of practical resolutions used in video generation policy. We agree that testing on higher resolution models further strengthen the analysis. However, due to the limitation of compute resource, we cannot reproduce Vidar which needs 64 NVIDIA Ampere-series 80GB GPUs. And high resolution generation leads to much higher latency, which is impractical for robotics tasks which require real time response.
>
> W2
>
> >Although the paper presents detailed algorithmic formulations, it omits an analysis of time complexity, particularly regarding the impact of token chunking. This makes it difficult for readers to understand the expected acceleration ratio or to choose suitable hyperparameters
>
> We analyze the time complexity based on function calls. Assume the cost of one model function call is $c$. In DDIM, the model is called $K$ times, so the baseline cost is $C_{\text{baseline}} = Kc$. In DTS, the draft sampling with step count $n_1$ has a cost $C_d = n_1 c$. When the chunk length is $L$, the draft cost becomes $C_d' = Lc$. The target sampling with step count $n_2$ has a cost $C_t' = n_2 c$. Under a token acceptance rate $\alpha$, the expected cost ratio is:
>
> $$
> \mathbb{E}\left[\frac{C_{\text{baseline}}}{C_{\text{DTS}}}\right]
> = \frac{K L \alpha}{n_1 L + n_1 n_2 }.
> $$
>
> However, in practice the cost of one target function call depends on the batch size, which corresponds to the chunk length $L$. If we ignore GPU parallelism and roughly assume that one target call costs $L$ times more, then the cost becomes $C_t' = L n_2 c$. In this case, the expected cost ratio simplifies to:
>
> $$
> \mathbb{E}\left[\frac{C_{\text{baseline}}}{C_{\text{DTS}}}\right]
> = \frac{K \alpha}{n_1 + n_1 n_2 }.
> $$
>
> We consider that increasing the chunk length can reduces NFEs and decoding time. Due to the GPU compute capacity, the increase of chunk length will reach the compute bound. Therefore, we except a relatively small chunk length and product of $n_1$ and $n_2$. We also tested different ratios under a chunk length of 6 on Thor dataset, and the experimental results align with the theoretical analysis.
>
> | **Ratio**      | **n1=20,n2=5** | **n1=20,n2=3** | **n1=20,n2=2** | **n1=8,n2=5** | **n1=10,n2=5** | **n1=12,n2=3** |
> |------------------|----------------|----------------|----------------|---------------|----------------|----------------|
> | **Overall SR**   | 27.9           | 17.1           | 21.7           | 30.4          | 29.15          | 23.0           |
> | **Speedup**      | 0.7418×        | 1.4990×        | 1.9289×        | 2.0539×       | 2.1445×        | 2.6570×        |
>
> W3
>
> >If the authors claim that generation is memory-bound, then the memory cost of token chunking (e.g., the number of model parameter loads) should be examined, as it may remain similar or even increase with chunking.
>
> We appreciate the reviewer’s comment. In Line 48, we claim that SD in LLMs can transform the decoding process from memory bandwidth bound to compute bound. We do not claim that visual generation is memory-bound. And our motivation is high resolution visual generation usually reaches GPU compute bound, while low-resolution video generation may not fully utilize them. In this setting, the goal of our method is not to reduce memory loads but to better balance compute usage across draft and target sampling when dealing with diffusion trajectories. Therefore, the memory cost of chunking does not affect the core motivation of our design.
>
> W4
>
> >The sequential nature of token chunking may limit parallelism. This trade-off should be discussed explicitly.
>
> We analysis chunk length in W2.The step sizes in a single draft sampling (chunk length) is the same as the batch size used for parallel target sampling (see Tables 4 and 5 in Appendix D.1). A smaller chunk length usually leads to faster sampling, while a larger chunk length often introduces redundancy and extra computation because of error accumulation, as described in Sec. 4.2. Another important factor for the sampling speed is the acceptance threshold (see Tables 7 and 8 in Appendix D.2). A smaller acceptance threshold requires stricter matching between draft and target tokens, which leads to rejection more likely and therefore increases the times of resampling.

---

> > ### Author Response · Authors · 2025-11-21
> > **Response to Reviewer pqge (2/3)**
> >
> > Q1
> >
> > >What is the motivation and intuition behind the Progressive Acceptance Strategy?
> >
> > The first motivation can be seen in line 248-251. Alignment between a coarse-grained and a fine-grained denoising trajectory is difficult as they are continues space. We proposed to relax the acceptance bound instead of strict matching to reduce resample.
> >
> > The second motivation can be seen in line 227: As both two sampling skip steps to produce the trajectory, the error is accumulated. Although the target sampling aims to achieves a better approximation and mitigate the problem of error accumulation, there still accumulate error. If we use strict matching or a fixed length relaxed bound, the algorithm will stop after the error exceeds the bound. We proposed PAS to progressively relax the acceptance bound to let the algorithm goes on.
> >
> > Q2
> >
> > >How does this work compare to Accelerated Diffusion Models via Speculative Sampling [2], which also avoids training a separate draft model?
> >
> > We propose a new method that utilize two complementary denoising trajectories based on the feature of ODE like DDIM, which did not exist in prior work on LLM SD or previous diffusion sampling. While [1] also proposed a training-free method, they focused on SDE. And they mentioned that “Speculative sampling is not applicable to deterministic samplers” in the conclusion, our method challenged this claim to make it possible.
> >
> >  [1] Accelerated Diffusion Models via Speculative Sampling
> >
> > Q3
> >
> > >Why is DDIM-10 chosen as the baseline solver for diffusion sampling instead of more advanced solvers such as DPM-Solver?
> >
> > Because one of our draft sampling settings uses 10 steps(similar to DDIM-10), and the target sampling stage performs a refinement on top of these 10 draft steps. We tested DPM-Solver with 10 function evaluations on Libero dataset and compared with different settings of GCP model(a variant of AVDC). The results show that DDPM-100 and our method both achieve higher SR than DDIM-100, DDIM-10, and DPM-Solver-10. We consider that the SR is also influenced by the action extractor rather than just video model. As a result, the benefit of advanced diffusion solvers may not directly translate into higher SR.
> >
> > | **Task**              | **DDPM-100** | **DDIM-100** | **DDIM-10** | **DPM-Solver** | **+ Ours** |
> > |-----------------------|--------------|--------------|-------------|----------------|------------|
> > | put-red-mug-left      | **45**       | 30           | 40          | 30             | 30         |
> > | put-red-mug-right     | 40           | 40           | 25          | 40             | **40**     |
> > | put-white-mug-left    | 55           | 40           | 50          | 45             | **70**     |
> > | put-Y/W-mug-right     | 30           | **40**       | **40**      | 35             | 25         |
> > | put-choc-left         | **70**       | 50           | 45          | 50             | 65         |
> > | put-choc-right        | 80           | 70           | 85          | 80             | **90**     |
> > | put-red-mug-plate     | 65           | **70**       | 50          | 55             | 65         |
> > | put-white-mug-plate   | 15           | 15           | 25          | 5              | **15**     |
> > | **Overall**           | 50           | 44.375       | 45          | 42.5           | **50**     |
> > | **Time cost (s)**     | 10.8335      | 10.8089      | 1.002       | 1.004          | **6.7578** |
> > | **Speedup**           | 1×           | 1.0022×      | 10.8119×    | 10.7903×       | **1.6031×** |
> >
> > Q4
> > >Could the proposed approach be extended to general image or video generation models, beyond policy-oriented settings?
> >
> > We thank the reviewer for this suggestion. (See line 48) Our work focuses on video generation policy as our motivation is that most visual generation tasks are compute bound as they generate high resolution images, while in visual generation for robotics, the generated content usually has a relatively low resolution. In principle, our method could be extended to other image or video generation task as it’s a training-free sampler.

---

> > > ### Author Response · Authors · 2025-11-21
> > > **Response to Reviewer pqge (3/3)**
> > >
> > > Q5
> > > >Why does DTS improve policy performance in addition to acceleration?
> > >
> > > Unlike sequential sampling, our method first produces a coarse global trajectory and then refines it. From another perspective, each point on the trajectory is effectively denoised twice, a coarse denoising followed by a fine denoising, and we only continue the process when these two estimates remain sufficiently consistent. In contrast, sequential sampling performs only a single denoising update at each step. We consider this consistency check leads to better policy performance in addition to faster sampling.
> > >
> > > >Were the results averaged over multiple runs to ensure statistical significance?
> > >
> > > We follow original setting in AVDC paper and report the mean SR. As for iThor, we report the mean SR aggregated from 3 types of objects per room with 20 episodes per object. As for Meta-world, we report the mean SR across tasks from 3 camera poses with 25 seeds for each camera pose.

---

> ### Author Response · Authors · 2025-11-26
> **Follow-up on Response**
>
> Dear Reviewer pqge,
>
> I hope this message finds you well. Your comments and the provided insights are very valuable to our work. As the discussion period is nearing its end with a week remaining, could you please review our response to see if it addresses your concerns? Your timely feedback will be extremely valuable to us. Could you read and let us know if there are more questions? We're eager to address any remaining questions to improve our work, and we hope that you will consider the significant contribution of this research to the field of video generation policy.
>
> Thanks for your time and effort in reviewing our paper.
>
> Regards,
>
> All the authors

---

### Official Review · Reviewer_yNA5 · 2025-10-29

**Soundness:** 3
**Presentation:** 3
**Contribution:** 2
**Rating:** 6
**Confidence:** 3

**Summary:**

In this paper, the authors focus on efficient video generation for robot policies. One major issue with contemporary video models is their slow speed. To address this issue, the authors propose the application of Draft-and-Target sampling, with some additional tweaks (chunking and progressive acceptance strategy) to video generation. They apply their approach to three datasets. They are iThor, Meta-World, and Libero, and they show improvements when accounting for the significant speedup in computational time.

**Strengths:**

1. The paper touches on an overlooked but very important problem with the current paradigm for video generation. It is generally too slow to be useful for robotics policies and planning algorithms. This paper is a promising step in the right direction.

2. The approach is simple and easy to implement.

3. Quantitative performance relative to baselines is promising.

**Weaknesses:**

1. The technical novelty is somewhat limited. No new models or approaches seem to be proposed in this paper. It appears that the contribution of this paper is largely an application of an existing idea to the realm of robotic control.

2. While performance is promising, the speedup is generally modest (about 2x). It is not clear if this speedup outweighs the additional complexity of the approach.

3. Data domain of video generation is quite constrained (robotic environments), there are no experiments on unconstrained video data (such as Kinetics-700)

**Questions:**

1. Could you please elaborate on the technical novelty of the approach? This seems to be an application paper. Are there any new methods or approaches?

2. The speedup from this approach is helpful but not particularly dramatic. Are there any tweaks to the method that could lead to additional speedup (> 5x) without significant sacrifices to performance?

3. Could this approach be viable for video generation on more diverse datasets beyond robotics?

---

> ### Author Response · Authors · 2025-11-21
> **Response to Reviewer yNA5**
>
> We thank reviewer for their careful and constructive review. Please also see our response to each of the questions below.
>
> Q1
>
> >Could you please elaborate on the technical novelty of the approach? This seems to be an application paper. Are there any new methods or approaches?
>
> We sincerely thank the reviewer for the detailed comments. Our work is not SD applied to robotic; instead, it introduces a new inference paradigm different from speculative decoding. We clarify the difference and novelty below.
>
> First, we work on continues state-space instead of discrete token such as words in LLM. LLM SD has well-defined probability distributions as LLMs produce common words or punctuation. In contrast, diffusion-based video policies focus on continuous denoising trajectories without a token-level probability distribution. This makes direct implement of SD to other domain inapplicable, which is consistent with the findings in [1]Sec 3.2 and [2]Sec 4.2.
>
> [1] Accelerated Diffusion Models via Speculative Sampling
>
> [2] Spec-VLA: Speculative Decoding for Vision-Language-Action Models with Relaxed Acceptance
>
> Second, most SD need to train an independent draft model. But our method is training-free as we propose a new method that utilize two different step size denoising trajectories based on the feature of ODE like DDIM. The combination of two complementary trajectories is a new construction which did not exist in prior work on LLM SD or previous diffusion sampling. Although [1] also proposed a training-free method, they focused on SDE. And they mentioned that “Speculative sampling is not applicable to deterministic samplers” in the conclusion, but our method challenged this claim to make it possible.
>
>  [1] Accelerated Diffusion Models via Speculative Sampling
>
> Q2
>
> >The speedup from this approach is helpful but not particularly dramatic. Are there any tweaks to the method that could lead to additional speedup (> 5x) without significant sacrifices to performance?
>
> We thank the reviewer for the suggestion. We would like to clarify that our method is orthogonal to several well-established acceleration techniques in diffusion models, which means additional speedup is indeed feasible.
>
> First, our baseline models produce denoising steps in pixel space, and prior work such as LDM has shown that moving from pixel space to latent space can achieve further acceleration while maintaining image quality. Our method can be directly applied in latent space, and this shift alone would compound with our method.
>
> Second, since our method is a training-free sampler. It can be inserted into more advanced DiT model and combine with other acceleration methods such as flash attention.
>
> Q3
> >Could this approach be viable for video generation on more diverse datasets beyond robotics?
>
> We thank the reviewer for this suggestion. (See line 48) Our work focuses on video generation policy as our motivation is that most visual generation tasks are compute bound as they generate high resolution images, while in visual generation for robotics, the generated content usually has a relatively low resolution. In principle, our method could be extended to other image or video generation task as it’s a training-free sampler.

---

> ### Author Response · Authors · 2025-11-26
> **Follow-up on Response**
>
> Dear Reviewer yNA5,
>
> I hope this message finds you well. Your comments and the provided insights are very valuable to our work. As the discussion period is nearing its end with a week remaining, could you please review our response to see if it addresses your concerns? Your timely feedback will be extremely valuable to us. Could you read and let us know if there are more questions? We're eager to address any remaining questions to improve our work, and we hope that you will consider the significant contribution of this research to the field of video generation policy.
>
> Thanks for your time and effort in reviewing our paper.
>
> Regards,
>
> All the authors

---

### Official Review · Reviewer_qRTA · 2025-11-01

**Soundness:** 2
**Presentation:** 2
**Contribution:** 2
**Rating:** 2
**Confidence:** 3

**Summary:**

The paper proposes a draft-and-target sampling method for video generation policy inference which achieves computational efficiency compared to prior works across benchmarks.

**Strengths:**

* The paper proposes system level optimization, e.g., token chunking, to improve the efficiency, which could offer practical values to the community.
* Results seem to provide empirical performance and efficiency gains.

**Weaknesses:**

* The core idea is the speculative decoding, which has been widely adopted in the field in LLMs. The modifications of speculative decoding in the discrete space from this paper include using large-stepsize-ODE as draft model and progressive acceptance, which are rather straightforward implementations. Therefore the contribution of the paper should be more explicitly discussed compared to prior works including but not limited to LLMs.
* Given that the paper uses the same model as draft and target models, these models have the same FLOP count per function evaluation. Reporting NFEs in the experiments could help clarify the computation complexity of the model.
* Baselines include AVDC (from the original paper cited) and AVDC-10 which aggressively cuts down denoising steps. More baselines with some numbers of denoising steps in between 10 and 100 should be reported, which might already achieve significant speedup compared to AVDC-100 without too much performance loss.

**Questions:**

* The experiments use AVDC as the backbone. Would the proposed strategy apply to other video generation policy models?

---

> ### Author Response · Authors · 2025-11-21
> **Response to Reviewer qRTA (1/2)**
>
> We thank reviewer for their careful and constructive review. Please also see our response to each of the questions below.
>
> W1
>
> >The core idea is the speculative decoding, which has been widely adopted in the field in LLMs. The modifications of speculative decoding in the discrete space from this paper include using large-stepsize-ODE as draft model and progressive acceptance, which are rather straightforward implementations. Therefore the contribution of the paper should be more explicitly discussed compared to prior works including but not limited to LLMs.
>
> We sincerely thank the reviewer for the comments. However, we disagree with the claim that our method is a straightforward implementations of SD. Our work is not SD applied to diffusion models; instead, it introduces a new inference paradigm different from SD. We clarify the difference below.
>
> First, we work on continues state-space instead of discrete token such as words in LLM. LLM SD has well-defined probability distributions as LLMs produce common words or punctuation. In contrast, diffusion-based video policies focus on continuous denoising trajectories without a token-level probability distribution. This makes direct implement of SD to other domain inapplicable, which is consistent with the findings in [1]Sec 3.2 and [2]Sec 4.2.
>
> [1] Accelerated Diffusion Models via Speculative Sampling
>
> [2] Spec-VLA: Speculative Decoding for Vision-Language-Action Models with Relaxed Acceptance
>
> Second, most SD need to train an independent draft model. But our method is training-free as we propose a new method that utilize two different step size denoising trajectories based on the feature of ODE like DDIM. The combination of two complementary trajectories is a new construction which did not exist in prior work on LLM SD or previous diffusion sampling.
> Although [1] also proposed a training-free method, they focused on SDE. And they mentioned that “Speculative sampling is not applicable to deterministic samplers” in the conclusion, but our method challenged this claim to make it possible.
>
>  [1] Accelerated Diffusion Models via Speculative Sampling
>
> W2
>
> >Given that the paper uses the same model as draft and target models, these models have the same FLOP count per function evaluation. Reporting NFEs in the experiments could help clarify the computation complexity of the model.
>
> We thank the reviewer for this suggestion. We conduct additional experiments on Thor dataset which shows the NFEs. The results are reported below. During policy rollout, the agent may occasionally get stuck and resample, resulting in different NFEs. Therefore, we record the total NFEs during the rollout. However, since the draft and target models have different costs per function call, we consider the sampling time/speedup is a more faithful metric.
>
> | **Method**    | Kitchen | Living Room | Bedroom | Bathroom | **Overall** | **Speedup** | **NFEs** |
> |---------------|---------|-------------|---------|----------|--------------|-------------|----------|
> | **AVDC-100**  | 30      | 16.7        | 35      | 26.7     | 27.1         | 1×          | 84800    |
> | **AVDC-50**   | 21.7    | 21.7        | 25      | 20       | 22.1         | 2.0154×     | 45950    |
> | **AVDC-25**   | 26.7    | 21.7        | 23.3    | 18.3     | 22.5         | 3.9593×     | 22525    |
> | **AVDC-10**   | 25      | 11.7        | 30      | 21.7     | 22.1         | 8.8358×     | 9380     |
> | **+ Ours**    | 30      | 25          | 38.3    | 23.3     | 29.15        | 2.1445×     | 19896    |
>
> We can see that our method used four times less NFE than DDIM-100 and achieved a higher SR

---

> > ### Author Response · Authors · 2025-11-21
> > **Response to Reviewer qRTA (2/2)**
> >
> > W3
> >
> > >Baselines include AVDC (from the original paper cited) and AVDC-10 which aggressively cuts down denoising steps. More baselines with some numbers of denoising steps in between 10 and 100 should be reported, which might already achieve significant speedup compared to AVDC-100 without too much performance loss.
> >
> >  We thank the reviewer for this suggestion. We conduct additional experiments on AVDC with different denoising steps(10, 25, 50, 100). The results are reported below. We find that the results of 10/25/50 steps are poor(with 22% SR), while 100 steps achieves 27.1% SR and our method with different n1 n2 ratio(n1 = 8, n2 = 5 and threshold = 12) achieves 30.4% SR.
> >
> > | **Method**       | Kitchen | Living Room | Bedroom | Bathroom | **Overall** | **Speedup** |
> > |------------------|---------|-------------|---------|----------|-------------|-------------|
> > | **AVDC-100**     | 30      | 16.7        | 35      | 26.7     | 27.1        | 1x          |
> > | **AVDC-50**      | 21.7    | 21.7        | 25      | 20       | 22.1        | 2.0154x     |
> > | **AVDC-25**      | 26.7    | 21.7        | 23.3    | 18.3     | 22.5        | 3.9593x     |
> > | **AVDC-10**      | 25      | 11.7        | 30      | 21.7     | 22.1        | 8.8358x     |
> > | **+ Ours**       | 33.3    | 18.3        | 38.3    | 31.7     | 30.4        | 2.0539x     |
> >
> > Q1
> >
> > >The experiments use AVDC as the backbone. Would the proposed strategy apply to other video generation policy models?
> >
> > We also test our method on GCP[1], which is a variant of AVDC. The results are reported in our original paper (Table 3, 6, 8). Our method also achieves up to 2.1x speedup with minimal compromise to the success rate.
> >
> > [1] GROUNDING VIDEO MODELS TO ACTIONS THROUGH GOAL CONDITIONED EXPLORATION

---

> ### Author Response · Authors · 2025-11-26
> **Follow-up on Response**
>
> Dear Reviewer qRTA,
>
> I hope this message finds you well. Your comments and the provided insights are very valuable to our work. As the discussion period is nearing its end with a week remaining, could you please review our response to see if it addresses your concerns? Your timely feedback will be extremely valuable to us. Could you read and let us know if there are more questions? We're eager to address any remaining questions to improve our work, and we hope that you will consider the significant contribution of this research to the field of video generation policy.
>
> Thanks for your time and effort in reviewing our paper.
>
> Regards,
>
> All the authors

---

### Meta-Review · Area_Chair_VBhq · 2026-01-02

**Summary:**

The reviewers broadly acknowledge the practical importance of improving inference efficiency for video generation policies in robotics, and recognize the empirical gains achieved by the proposed DTS, particularly its training-free design and minimal performance degradation. However, significant concerns converge around limited technical novelty, inadequate baseline comparisons, and insufficient theoretical or systems-level analysis.

**Reviewer Concerns:**

The rebuttal addresses several concerns: new metrics (e.g., NFEs) and additional experiments (e.g., intermediate-step baselines) were provided, and the motivation for key components (e.g., progressive acceptance) was clarified, improving overall clarity.
However, two core issues remain:
- The novelty concern raised by reviewers qRTA and yNA5 persists. The rebuttal emphasizes distinctions from prior speculative decoding work but does not establish a fundamentally new algorithmic contribution.
- The viability beyond robotics, questioned by yNA5 and pqge, is still not empirically validated; the authors assert extendability “in principle,” but no experiments on non-robotic or high-resolution video data were added.

**Reviewer Scores:**

- Reviewer qRTA may increase the score, as the request of additional experiments and metrics is provided by the authors. (Original score: 2, After rebuttal: 2 or 4).
- Reviewer yNA5 is unlikely to increase the score, given persistent concerns about novelty and limited speedup magnitude. (Score: 6).
- Reviewer pqge may increase the score, as the questions about theoretical analysis is answered but may not be thorough enough. (Original score: 2, After rebuttal: 2 or 4).

---

### Decision · Program_Chairs · 2026-01-26

Reject